# PlantDeepMeth: A Deep Learning Model for Predicting DNA Methylation States in Plants

**DOI:** 10.3390/plants14111724

**Published:** 2025-06-05

**Authors:** Zhongwei Guo, Wenyuan Fan, Chengcheng Cai, Kang Zhang, Xilin Hou, Ying Li, Feng Cheng

**Affiliations:** 1National Key Laboratory of Crop Genetics and Germplasm Enhancement, College of Horticulture, Nanjing Agricultural University, Nanjing 210095, China; 82101199706@caas.cn (Z.G.); hxl@njau.edu.cn (X.H.); 2State Key Laboratory of Vegetable Biobreeding, Key Laboratory of Biology and Genetic Improvement of Horticultural Crops of the Ministry of Agriculture and Rural Affairs, Institute of Vegetables and Flowers, Chinese Academy of Agricultural Sciences, Beijing 100081, China; fan724963@126.com (W.F.); caichengcheng@caas.cn (C.C.); zhangkang01@caas.cn (K.Z.); 3Key Laboratory of Vegetable Biology of Hainan Province, Haikou 571100, China

**Keywords:** deep learning, convolutional neural networks, recurrent neural network, methylation state, plant epigenomics

## Abstract

Cytosine DNA methylation (5mCs) is an important epigenetic modification in genomic research. However, the methylation states of some cytosine sites are not available due to the limitations of different studies, and there are few tools developed to deal with this problem, especially in plants, which have more methylation types than animals. Here, we report PlantDeepMeth, a novel deep learning model that utilizes deep learning to predict DNA methylation states in plants. The evaluation of PlantDeepMeth on known cytosine sites in both the *Brassica rapa* and *Arabidopsis thaliana* genomes shows good performance in predicting methylation states, indicating that the tool is good at learning patterns for methylation imputation. Motif analysis of the model’s predictions identified specific motifs associated with hypo- or hyper-methylation states in *B. rapa* and *A. thaliana*, further revealing key regulatory patterns captured by the model. Moreover, cross-species validation between *B. rapa* and *A. thaliana* demonstrated the generalizability of PlantDeepMeth, with the model maintaining high performance across different plant species. These results highlight the effectiveness of PlantDeepMeth and demonstrate the potential of deep learning to advance plant genomics research.

## 1. Introduction

Cytosine DNA methylation (5mCs) is a critical epigenetic modification that is widely present in genomes of plants and animals and plays an important role in regulating gene expression, maintaining genome stability, and influencing developmental processes [1,2,3,4,5,6,7]. With the advent of high-throughput sequencing technologies, vast amounts of DNA methylation data have been generated, providing essential resources for in-depth studies of DNA methylation functions in various biological activities. However, due to technical limitations of specific experiments such as single-cell methylation sequencing, and biological complexities such as high GC content and repetitive genomic regions, and those that show structural variation between the studied genome and the reference genome, as well as those with insufficient sequencing depth, methylation data often suffer from uneven coverage [8,9,10]. These limitations result in lost methylation information or inaccurate determination of methylation states in certain genomic regions.

The presence of incomplete or low-quality methylation sites limits the study of genome-wide methylation patterns. To overcome this problem, researchers have proposed various methods to impute or correct the methylation states of these sites [11,12,13]. These methods typically leverage genomic features such as CpG island proximity, transcription factor binding sites, histone modification marks, and DNA sequence context to improve prediction accuracy. Nevertheless, these approaches often struggle with accuracy and generalization, especially when dealing with large, complex methylation data. Furthermore, while these methods benefit from the comprehensive genomic annotations available for humans and some animals, such annotation information is often lacking for plants, limiting the development of accurate methylation prediction models specifically for plant genomes.

In recent years, deep learning has emerged as a powerful data-driven approach that breaks the limitations of traditional methods (here referred to as machine learning) by reducing the reliance on well-annotated genomes [14,15,16,17,18,19,20]. Deep learning models can learn and extract high-level features directly from raw data, enabling accurate prediction of intricate biological patterns even in the absence of genomic annotation data. For instance, DeepCpG is a deep learning-based model that uses only DNA sequences and neighboring methylation data to predict methylation states in human and animal single cells, outperforming traditional methods such as Random Forest (RF) that rely on multiple genomic annotation information [21]. This data-driven approach has potential for plant genomes, which often lack detailed genomic annotations. However, because plant genome sequences have three types of methylation (CpG, CHG, and CHH, where H = A, C, or T), traditional methods and existing deep learning models, such as DeepCpG, were developed for animal genomes with only CpG sites, and thus cannot handle the complexity and diversity of methylation patterns in plants. The three types of methylation characteristic of plant genomes make it impossible to directly apply deep learning models developed for animal genomes to plants. In addition, the Smart Model for Epigenetics in Plants (SMEP) shows the potential of deep learning to predict diverse epigenomic modifications in plant genomes using DNA sequences. However, relying solely on DNA sequence information may not comprehensively capture the intricate patterns associated with plant DNA methylation. Other models, such as MethSemble-6mA, iResNetDM, and Methyl-GP, leverage ensemble learning or sequence-based representations to identify DNA and RNA methylation sites across different animal species [22,23,24]. Furthermore, frameworks like DeepPlantCRE and StableDNAm have integrated attention mechanisms, residual networks, and contrastive learning to enhance feature learning in plant regulatory regions [25,26]. These advances reflect the growing recognition that plant methylomes require specialized architectures to address their unique biological and structural features.

Transfer learning has emerged as a promising approach to overcome this challenge. Transfer learning is a deep learning method that exploits the knowledge learned from one domain (the source domain) and applies it to a different but related domain (the target domain), enabling the model to perform well in the target domain even with limited labeled data by utilizing the abundant labeled data from the source domain. It is a particularly suitable approach to apply models trained in animal genomes to plant genomes. For example, DeepSEA, a deep learning model originally developed to predict the functional effects of non-coding variants in the human genome [27], has been successfully transferred to predict the regulatory effects of genomic variants in plant genomes [28]. This demonstrates the use of transfer learning to bridge the gap between the distinct genomic features of animals and plants, and provides a viable route for the efficient development of plant genome models. Another example is DeepSignal-Plant, which was transferred from DeepSignal. It relies on third-generation sequencing data to make cytosine methylation state predictions. But DeepSignal-Plant cannot predict cytosine methylation states for regions lacking sequencing signal coverage, as it depends on signal-based features. In addition, DeepPlant was proposed as an enhanced model for plant methylation detection, which is also based on third-generation sequencing data. It combines BiLSTM and Transformer-based triple encoders to better capture methylation patterns, particularly for underrepresented CHH sites [29].

With the aim of predicting/correcting the methylation states in plant genome studies, we developed PlantDeepMeth, a transfer learning model based on DeepCpG, which was originally designed for animal genomes and adapted for plant genomes. Because the methylation pattern of plants is different from that of animals, we modified the structure of DeepCpG and retrained the entire network from scratch using the plant methylation data. The results showed that PlantDeepMeth can effectively predict the methylation states of cytosine sites in plant genomes and find motifs associated with different patterns of methylation activity. Overall, PlantDeepMeth provides an effective solution to the problem of predicting DNA methylation in plants, supporting the power of deep learning and transfer learning in bioinformatics.

## 2. Materials and Methods

### 2.1. Data Collection and Process

The bisulfite sequencing dataset focusing on cytosine DNA methylation (5mCs) from leaves of *Brassica rapa* under normal growth conditions was retrieved from NGDC under accession numbers CRR596509 [30]. The reference genome (V3.0) and annotation datasets (V3.1) of *B. rapa* were downloaded from http://brassicadb.cn/#/Download/ (accessed on 31 July 2024) [31]. The bisulfite sequencing of wild-type *Arabidopsis thaliana* was retrieved from NCBI under accession numbers SRR15967549 [32]. The reference genome (TAIR10) was downloaded from http://plants.ensembl.org/index.html (accessed on 21 August 2024) [33]. The bisulfite sequencing of *Oryza sativa* was obtained from NCBI under accession numbers SRR1542709. The *O*. *sativa* reference genome (IRGSP-1.0) was downloaded from the Ensemble Plants database (https://plants.ensembl.org/index.html, accessed on 19 September 2024).

Bisulfite sequencing data were aligned to the reference genome using Bismark (v0.24.2) [34]. After alignment, methylation calls were extracted for each cytosine site. A custom Perl script was employed to count the number of reads at each cytosine site; only cytosine sites with at least four aligned reads were used for training. Cytosine sites with fewer than four reads were labeled as ‘NA’ and excluded from the training process (Figure 1a, ‘?’ represents ‘NA’). Therefore, the positive samples (methylation sites) and negative samples (unmethylation sites) were generated from the bisulfite sequencing data. After model training, these ‘NA’ values were filled based on the model’s predictions. The input methylation status was rounded according to the methylation rate, with the values of 0 and 1 representing unmethylated and methylated states, respectively (Figure 1a). For *B. rapa*, the data from chromosomes 1 to 7 were used as the training set, while data from chromosomes 8 and 9 were used as the validation set. The data from chromosome 10 was reserved as the testing set. Similarly, for *A. thaliana*, the data from chromosomes 1 to 3 were used for training, data from chromosome 4 for validation, and data from chromosome 5 for testing. This partitioning strategy ensures that the model is evaluated on unseen data from different chromosomes, allowing for a more robust assessment of its generalization performance across the genome. Models were initially fitted on the training sets, and the validation sets were used to optimize parameters. The final model performance was evaluated on the testing sets. For computing binary evaluation metrics, such as AUC, F1 score, or MCC score, predicted methylation probabilities greater than 0.5 were rounded to 1 (methylated), while those less than or equal to 0.5 were rounded to 0 (unmethylated).

### 2.2. PlantDeepMeth Model Architecture

The PlantDeepMeth model was transferred from DeepCpG [21], specifically adapted for plant genomes to predict cytosine methylation states. Unlike DeepCpG, which uses single-cell methylation sequencing as input, PlantDeepMeth integrates three methylation types of cytosine sites from plant genomes, allowing the model to learn broader methylation patterns and capture more generalizable features. Due to the distinct methylation contexts in plant genomes as compared to animals, we did not freeze any layers from the original DeepCpG. Instead, we modified the model architecture and retrained all layers from scratch using plant methylation data. The PlantDeepMeth model consists of three distinct models: a DNA model that captures and learns features from DNA sequences, a methylation model that focuses on extracting features from the surrounding regions of cytosine sites, and a joint model that integrates the learned features from both the DNA and methylation models to make comprehensive predictions. All models were built in Python 3.6 using Keras (version 2.2.5) with TensorFlow (version 1.14) as the backend. All models were trained and evaluated on a Linux server equipped with an Intel (R) Xeon (R) Gold 5120 CPU @ 2.20 GHz, 320 GB RAM, and an NVIDIA Tesla V100-PCIE-32 GB GPU. The system ran Linux kernel (version 3.10.0-1160.el7.x86_64) compiled with GCC 7.3.0, using CUDA version 10.1.

### 2.3. DNA Model

The DNA model in PlantDeepMeth is designed as a convolutional neural network (CNN) to effectively capture and process sequence features from a 1001 bp DNA sequence centered on a target cytosine site (500 bp upstream and downstream of the target central site). The DNA sequence was represented as a binary matrix using one-hot encoding for the four nucleotides: A = [1, 0, 0, 0], T = [0, 1, 0, 0], G = [0, 0, 1, 0], and C = [0, 0, 0, 1]. The architecture of the DNA model processes the input DNA sequence through multiple convolutional layers, which are constructed with two convolutional layers. The first convolutional layer uses 128 filters with a kernel size of 11 and a stride of 1, followed by max pooling with a pool size of 4. The second convolutional layer uses 256 filters with a kernel size of 3 and a stride of 1, followed by max pooling with a pool size of 2. Each convolutional layer is followed by a ReLU activation function. To mitigate the risk of overfitting and improve the model’s generalization ability, a dropout layer is applied after the fully connected layer, randomly omitting a portion of neurons during training. The features extracted by the convolutional layers are then flattened into a 1D vector, which is fed into a fully connected layer with 128 units activated by ReLU. The final output of the DNA model is generated by a predictive layer that uses a sigmoid activation function.

### 2.4. Methylation Model

The methylation model in PlantDeepMeth is designed to capture the complex relationships and dependencies between methylation states of neighboring cytosine sites. This model employs two layers of bidirectional Gated Recurrent Units (GRUs) to process the sequential nature of methylation data. The input of this model consists of 100-dimensional vectors that encode the methylation states and distances to the 25 neighboring cytosine sites on both sides of a central cytosine site, with the methylation state of the central site being used as the label. These distances are normalized by dividing them by the maximum genome-wide distance (i.e., the largest possible distance between any two neighboring cytosine sites across the entire genome), transforming them into relative ranges that allow the model to effectively learn positional relationships across different scales.

Initially, the input sequences are merged and processed by a Time Distributed layer, which applies the same replicated model to each time step independently (i.e., the model is applied separately to each position in the sequence without considering interactions between different time steps). This is followed by the first bidirectional GRU layer with 128 units, where both L1 and L2 regularizations were applied with coefficients of 0.0001 to reduce overfitting and promote generalization to unseen data. The second bidirectional GRU layer, with 256 units, further refines the feature representation by capturing more complex temporal dependencies. A final dropout layer is used to further mitigate overfitting by randomly omitting a fraction of GRUs during training. The output is a 512-dimensional vector that captures complex temporal dependencies in methylation state transitions around the target cytosine.

### 2.5. Joint Model

The joint model in PlantDeepMeth takes as input the concatenated feature vectors from the DNA and methylation models. Specifically, it receives a 768-dimensional input vector formed by combining the 256-dimensional output from the DNA model and the 512-dimensional output from the methylation model. The combined feature vector is then passed through two fully connected layers, each with 512 units and ReLU activation. The output layer consists of a final dense layer with a sigmoid activation function, which is used to predict the binary methylation state (methylated or unmethylated) for each cytosine site. The joint model effectively integrates the features learned from the DNA sequence and local methylation context. By modeling the interactions between the extracted DNA sequence and surrounding cytosine features, the model can capture higher-order dependencies that might be missing when considering each feature set independently.

### 2.6. Model Evaluation

In this study, we evaluate the performance of PlantDeepMeth models using the following key metrics: accuracy (ACC), area under the receiver operating characteristic curve (AUC), F1 score, Matthew’s correlation coefficient (MCC), true negative rate (TNR), and true positive rate (TPR). The default threshold for all the evaluation indicators is 0.5. ACC provides a general measure of the model’s performance by calculating the ratio of correct predictions to the total number of predictions. However, given the class imbalance in our dataset, i.e., the proximity ratio between methylated and unmethylated sites is 1:9 in *B. rapa* and *A. thaliana*. AUC is particularly valuable in this context as it reflects the model’s ability to distinguish between positive and negative classes, regardless of the threshold. The F1 score, which is the harmonic mean of precision and recall, offers a balanced measure, especially when dealing with uneven class distributions. MCC, designed for binary classification, provides a more informative metric than accuracy in cases of class imbalance by considering all four confusion matrix categories (true positives, true negatives, false positives, and false negatives). TNR and TPR are used to evaluate the model’s performance in identifying negative and positive instances, respectively.

### 2.7. Motif Analysis

The motif analysis was performed using the DNA model trained on *B*. *rapa* and *A. thaliana*. Specifically, we used the kernels from the first convolutional layer of the DNA model. Each kernel in this layer has a length of 11 base pairs. This fixed kernel size allows the model to capture specific sequence patterns of this length, which are then interpreted as potential motifs associated with DNA methylation.

The filters from the convolutional layer of the DNA model were first analyzed to identify the sequence fragments that produced the highest activation values. These sequence fragments were then aligned to visualize the patterns captured by each filter. The activation levels of all filter neurons were computed for a set of DNA sequences. For each sequence and filter, a sequence window around the position that activated the filter the most was selected. If the activation level of the filter at a specific position was greater than 50% of the maximum activation level across all sequences, the corresponding sequence window was selected. These selected sequence windows were then aligned and visualized as sequence motifs using WebLogo (version 3.7.8) [35].

## 3. Results

### 3.1. The Construction and Workflow of PlantDeepMeth

We developed PlantDeepMeth to integrate a convolutional neural network (CNN) for DNA sequence processing and a recurrent neural network (RNN) for capturing sequential dependencies within cytosine DNA methylation data (Figure 1a). PlantDeepMeth implemented three models: DNA model, methylation model, and joint model. Using these three models, PlantDeepMeth can infer lost or correct inaccurate methylation states often found in genomic regions with no/low coverage of DNA methylation sequencing reads, filling the gap in DNA methylation prediction tools in plant genomic studies (Figure 1b). Rather than relying on increasing sequencing depth, which can be prohibitively costly and not always feasible in plant genome studies, PlantDeepMeth offers a more cost-effective solution.

In detail, the DNA model processed 1001 bp sequences (500 bp upstream and 500 bp downstream of the central cytosine site) to extract relevant features. It used a CNN to identify patterns in the DNA sequence, which were then passed through layers to extract and refine features for methylation state prediction. The methylation model input is a 100-dimensional vector, gathering methylation state and distance information from neighboring cytosine sites to predict the methylation state of targets. By exploiting the sequential nature of methylation data, it used recurrent layers that capture dependencies between the methylation states of nearby sites, allowing the model to account for the spatial relationships inherent in the DNA methylation process. The joint model integrated the outputs of the DNA and methylation models by concatenating the extracted features. This combined information was then passed through additional layers to model the interactions between the DNA sequence and local methylation signals, allowing prediction of the methylation state (Figure 1a).

For the running process, the bisulfite sequencing data were first aligned to the reference genome. Each cytosine site was then classified as either methylated or unmethylated based on the alignment of the sequencing reads at that site. If the total number of reads was less than four, the cytosine site was labeled as unknown (‘NA’ or ‘?’). The sequence information was then encoded into a binary matrix, representing the 1001 bp DNA sequence centered on the target unknown cytosine site. This encoded sequence was then processed by the CNN to capture key sequence features, while the methylation information from neighboring cytosine sites was processed by a bidirectional GRU network to model temporal dependencies. The joint model was further run based on the outputs of the DNA and methylation models. All three models were trained, verified, and evaluated after model architecture construction.

### 3.2. PlantDeepMeth Accurately Predicts DNA Methylation States

To evaluate the performance of PlantDeepMeth, we applied the model to predict observed methylation states in *A. thaliana* and *B. rapa* and compared the results with those obtained using SMEP, a published deep learning model for 5mC prediction in plants. Considering that positive samples (methylated cytosine sites) and negative samples (unmethylated cytosine sites) are not balanced in the datasets, accuracy in this case cannot truly reflect model performance, we used the area under the receiver operating characteristic curve (AUC) and the area under the Precision-Recall Curve (PRC) as key indicators to evaluate model performance. In addition to using AUC and PRC, our model was also evaluated using several performance metrics to gain a comprehensive understanding of its effectiveness. Figure 1c presents the results for *A. thaliana*, while Appendix A shows the evaluation on *B. rapa*.

The combined accuracy, AUC, and PRC values support that all three models in PlantDeepMeth capture distinct patterns associated with DNA methylation (Figure 1c and Appendix A). The performance highlights the robustness of PlantDeepMeth in distinguishing between methylated and unmethylated cytosine sites, with all three models achieving accuracy and AUC values above 0.8. Using the *A. thaliana* data as an example, the DNA model achieved an accuracy of 0.82, an AUC of 0.81 and a PRC of 0.15; the methylation model showed stronger performance with an accuracy of 0.86 and an AUC of 0.93 and a PRC of 0.49; while the joint model showed the best performance with an accuracy of 0.89 and an AUC of 0.95 and a PRC of 0.53 (Figure 1c). In contrast, SMEP, which relies solely on 41 bp DNA sequence inputs, achieved comparatively lower performance in *A. thaliana* with an accuracy of 0.66, an AUC of 0.50, and a PRC of 0.08. Similar results were obtained in *B. rapa* (Appendix A). Moreover, the joint model analysis in *A. thaliana* and *B. rapa* outperformed both the DNA and methylation models in various metrics (Figure 1c and Appendix A), indicating that the joint model can effectively integrate both DNA sequence and methylation information, learn high-level patterns, and thus provide a more comprehensive prediction framework. In addition, the consistent performance of PlantDeepMeth in both *A. thaliana* and *B. rapa* indicates its potential as a reliable tool for methylation prediction in diverse plant species (Figure 1c and Appendix A).

### 3.3. Identification of Motifs Associated with Cytosine Methylation

Sequence motifs that were associated with cytosine methylation in the *A. thaliana* and *B. rapa* genomes identified by the DNA model were analyzed. The DNA model was specifically trained to detect and characterize sequence motifs that could influence methylation states. The filters from the first convolutional layer of the DNA model, which function similarly to conventional position weight matrices, were visualized as sequence logos to facilitate interpretation of the motifs recognized by the model. Representative motifs from *B. rapa* are shown in Figure 2, while those from *A. thaliana* are presented in Appendix A.

Principal component analysis (PCA) was applied to the activity scores of these motifs, and the results show distinct patterns of motif co-occurrence and their estimated effect on methylation states. Specifically, the PCA plot shows that motifs with similar nucleotide composition tend to cluster together. As a result, some motif labels appear overlapped in the figure due to close proximity in the PCA space, but this does not affect the interpretation of clustering patterns. Two major clusters emerge: one associated with increased methylation levels and the other with decreased methylation levels (Figure 2 and Appendix A). Consistent with patterns observed in a deep learning model of animals [21], motifs associated with decreased methylation levels were often found to be CG-rich and predominantly distributed in the *B. rapa* and *A. thaliana* genomic regions with higher CG content, such as promoter regions and transcription start sites. These motifs are likely to be involved in regulatory processes that maintain hypomethylation in active promoter regions. On the other hand, motifs associated with increased methylation levels tended to be AT-rich and were more abundant in CG-poor genomic contexts, possibly indicating their role in establishing or maintaining hypermethylation in these genomic regions. These motifs could serve as molecular markers for future studies of the influence of DNA sequence on methylation patterns, as well as their broader role in regulating gene expression and plant development.

### 3.4. Cross-Species Prediction

Generalizability is a critical aspect of deep learning, and strong generalizability indicates that a model can apply learned knowledge to other datasets. To assess the generalizability/robustness of the PlantDeepMeth models, we performed cross-species prediction experiments between *B. rapa* and *A. thaliana*. Specifically, PlantDeepMeth models trained on one genome were used to predict methylation states in the other genome. Our results highlight the generalization ability of all three models in PlantDeepMeth, with the joint model showing the best performance (Figure 3). The learned patterns of methylation states in PlantDeepMeth are applicable to the other genomes of *B. rapa* or *A. thaliana*, suggesting that the methylation regulatory mechanism may be conserved between them. This cross-species evaluation underscores the potential of deep learning models to adapt to the unique epigenetic landscapes of different organisms, a key strength of deep learning methods in bioinformatics. These results are summarized in Appendix A.

Although PlantDeepMeth can predict missing methylation states, we cannot evaluate its prediction accuracy on these sites due to the lack of experimental verification. Using the PlantDeepMeth trained on *A. thaliana*, we can evaluate its performance on these known methylation states in *B. rapa.* The joint model achieved an AUC of 0.92 and an F1 score of 0.56, significantly outperforming the DNA model, which had an AUC of 0.74 and an F1 score of 0.39 (Figure 3a), and also better than the methylation model, which had an AUC of 0.88 and an F1 score of 0.44. The higher accuracy and Matthew’s correlation coefficient (MCC) of the joint model indicate its superior ability to capture methylation patterns across species. When we evaluated the performance of PlantDeepMeth trained on *B. rapa* for predicting methylation states in *A. thaliana*, the joint model again demonstrated its highest strength with an AUC of 0.92 and an F1 score of 0.49 (Figure 3b). In comparison, the DNA model showed an AUC of 0.71 and an F1 score of 0.16, while the methylation model achieved an AUC of 0.92 and an F1 score of 0.41. Furthermore, despite the lower F1 score in the DNA model, its true negative ratio (TNR = 0.91) was the highest among the models, highlighting its ability to correctly identify unmethylated states (Figure 3b). Interestingly, although the joint model generally outperformed the other models, the methylation model had a higher true positive ratio (TPR = 0.86 for *A. thaliana* predicting *B. rapa*, and 0.84 for *B. rapa* predicting *A. thaliana*), indicating its superior ability to identify methylated sites (Figure 3). This suggests that the methylation model, which focuses on local methylation signals, may be more sensitive in detecting true positives, even though the joint model captures broader patterns across species.

In addition, to better verify the generalization ability of PlantDeepMeth, we further applied the learned rules of PlantDeepMeth from the *A. thaliana* genomes to the *Oryza sativa* genome. The results showed that all three models performed well, with the joint model achieving the highest performance (Appendix A). This indicates that the ability of the model to capture methylation patterns is not limited to *B. rapa* and *A. thaliana* but has a broad applicability. Take together, these results highlight the generalizability and robust performance of PlantDeepMeth models, especially the joint model, across different plant species, demonstrating their effectiveness even when applied to diverse and phylogenetically distant plant genomes.

## 4. Discussion

In this study, we proposed PlantDeepMeth to address the challenge of missing or uncertain cytosine methylation states in plant genomes caused by uneven sequencing coverage and biological complexity. By modifying and retraining the DeepCpG architecture on plant-specific data, our model captures the distinct methylation types in plants—including CpG, CHG, and CHH, which are not addressed by existing models originally developed for animals. Evaluated across *B. rapa*, *A. thaliana*, and *Oryza sativa*, PlantDeepMeth shows strong cross-species generalizability and consistently outperforms SMEP, other deep learning models developed for plant methylation prediction from second-generation sequencing. Moreover, the model identifies sequence motifs associated with methylation variation, providing new insights into plant epigenetic regulation. These results demonstrate the advantages of PlantDeepMeth and highlight the potential of deep learning for advancing plant methylome research.

Although PlantDeepMeth performed well in our study, it does have some limitations. While the model generalizes well across closely related species, it may struggle with more distantly related species, which could require species-specific retraining to capture their unique epigenetic landscapes. Moreover, using *B. rapa* as an example, the current training dataset is derived exclusively from leaf tissues under standard growth conditions. As a result, the model may not generalize well to other tissues or to plants exposed to environmental stress. For instance, we observed notably reduced performance when applying the model to maize embryo tissues (Appendix A), suggesting that methylation patterns can differ substantially across developmental stages and physiological contexts. These findings highlight the need for future development of tissue-specific models to improve prediction accuracy across diverse biological contexts.

Another limitation of PlantDeepMeth is the lack of direct comparison with traditional machine learning methods that are typically used in other contexts, such as RF models. For example, the RF Zhang model leveraged a comprehensive set of features, including the methylation state and distance of neighboring CpG sites, annotated genomic contexts, transcription factor binding sites, histone modification marks, and DNase I hypersensitivity sites [36]. These features were derived from comprehensive genomic annotations. However, the RF Zhang model was originally developed for animal genomes, and as such, it does not contain the CHH and CHG methylation types. Meanwhile, such detailed genomic annotation information is currently not available for most plant genomes, limiting the direct comparison of our method with traditional machine learning methods. Beyond SMEP, several other methylation prediction methods have emerged, including those for RNA methylation, such as m5C site identification using XGBoost with feature selection [37]. Although effective in their domains, these models are typically trained on RNA or animal datasets and do not address the structural or epigenetic complexity of plant genomes. DeepSignal-Plant and DeepPlant support all three methylation types using Nanopore sequencing but focus on signal-level detection rather than imputation from sparse bisulfite data [29,32]. In contrast, PlantDeepMeth is specifically designed to address this challenge by integrating DNA sequence features and local methylation context to impute missing methylation states from plant bisulfite sequencing data.

The methylation model relies on methylation patterns derived from neighboring cytosine sites and works well mainly in genomic regions with sufficient and accurate methylation information. In certain genomic regions, such as telomeres or centromeres, where successive cytosine sites often have no or low read coverage, their methylation information is largely incomplete or inaccurate. In this situation, the site with reliable methylation data used by the methylation model for context may be too distant from the prediction site, leading to reduced prediction accuracy. In comparison, by utilizing DNA sequence features instead of relying exclusively on methylation patterns, the DNA and joint models are better at making accurate predictions in such genomic regions. In addition, although the joint model provides a more robust and reliable solution by combining sequence and methylation information, it tends to run slower due to its computational load.

The DNA model identified sequence motifs that appear to be associated with methylation activity in *B. rapa* and *A. thaliana*. These motifs provide valuable insights into the regulatory elements that may be marked by methylation dynamics. Specifically, we observed that certain motifs were consistently linked with regions of high methylation, suggesting a potential role in promoting or maintaining DNA methylation. Conversely, some other motifs were associated with regions exhibiting reduced methylation, possibly indicating a role in demethylation processes or the prevention of methylation in specific genomic contexts. In *B. rapa*, a non-model organism with limited prior data on DNA methylation motifs, these findings serve as a foundational reference for exploring its epigenetic landscape. In *A. thaliana*, a well-studied species, our results confirmed existing knowledge about methylation-related regulatory elements while also presenting new motifs for experimental validation. Future work can focus on validating the biological significance of these motifs, particularly their role in methylation regulation and interactions with other epigenetic factors such as histone modifications. Understanding these motifs in greater detail may reveal critical insights into methylation regulation dynamics and offer guidance for manipulating methylation patterns in plant breeding and functional genomics.

Future work will involve constructing tissue-specific models and adapting the model to account for dynamic methylation changes under stress or other conditions. Meanwhile, with the availability of third-generation sequencing data, it is expected that the deep learning approach will be extended to more accurately impute missing methylation states. In addition, further work needs to focus on de novo construction of models, refining the model architecture, optimizing training strategies, and integrating more genomic features to improve the prediction accuracy.

## Figures and Tables

**Figure 1 plants-14-01724-f001:**
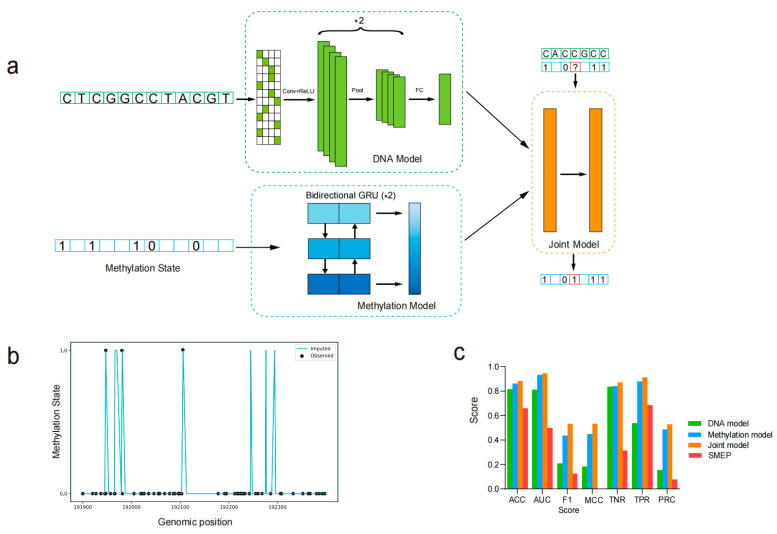
Overview and performance evaluation of the PlantDeepMeth model. (**a**) PlantDeepMeth architecture and workflow. The model consists of three models. The DNA model is a convolutional neural network that processes DNA sequences, extracting features through convolutional, pooling, and fully connected layers. The methylation model uses a bidirectional Gated Recurrent Unit to capture patterns from neighboring methylation states. Both models were fed into the joint model, which combines features from DNA sequences and methylation states through fully connected layers to predict the methylation state at each cytosine site. (**b**) Methylation state prediction on chromosome 10 of *Brassica rapa* using the joint model. Visualization of observed and imputed methylation states across a genomic region. The black dots indicate the observed methylation states, while the cyan line represents the imputed states from the model. (**c**) The performance evaluation of PlantDeepMeth trained on *Arabidopsis thaliana* and comparison with the SMEP model.

**Figure 2 plants-14-01724-f002:**
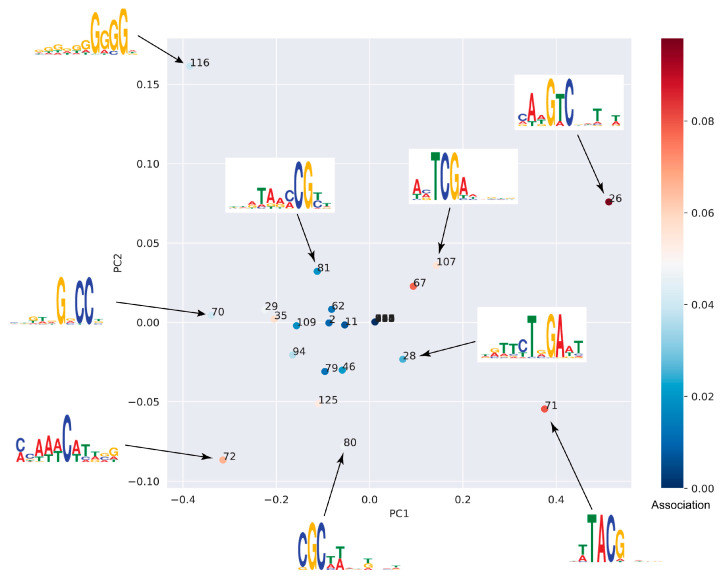
Discovered sequence motifs associated with DNA methylation in *Brassica rapa*. The motifs were identified by PlantDeepMeth after filtering partial sequences. The figure shows the first two principal components of the motif occurrence frequencies in sequence windows (activity). The low to high estimated motif effect on methylation levels is represented by blue to red colors. Sequence logos show the motif related to DNA methylation.

**Figure 3 plants-14-01724-f003:**
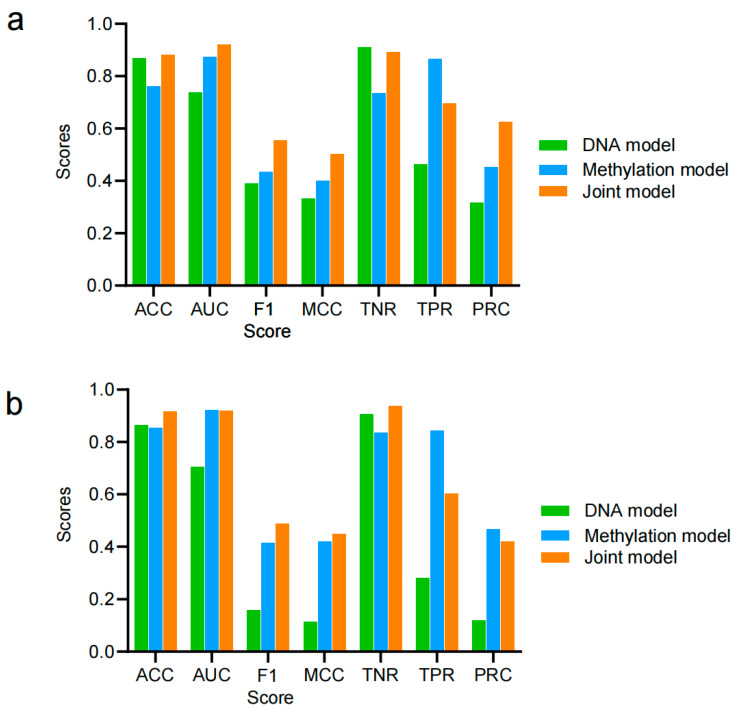
Cross-validation of PlantDeepMeth based on *Brassica rapa* and *Arabidopsis thaliana*. (**a**) The evaluation of PlantDeepMeth trained on *A. thaliana* for predicting *B. rapa*. (**b**) The evaluation of PlantDeepMeth trained on *B. rapa* for predicting *A. thaliana*.

## Data Availability

Data are contained within the article and Appendix A. The source code of PlantDeepMeth is freely available at: https://gitee.com/Bioinformaticslab/PlantDeepMeth (accessed on 21 August 2024).

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
