# Peer review of "PlantDeepMeth: A Deep Learning Model for Predicting DNA Methylation States in Plants"

_plants, 2025, doi:10.3390/plants14111724_

Round 1
Reviewer 1 Report
Comments and Suggestions for Authors
The manuscript presents a nice work however I have few queries before the acceptance of the manuscript,
@ The use of a 0.5 threshold for binary classification is standard but not always optimal. Indicate whether other thresholds were tested for best performance.
@ Reproducibility is necessary, the code is must and please specify whether random seeds were fixed and describe the hardware/software environment used for training.
@ Key parameters like convolutional filter size and stride are missing. These are important for understanding how the model processes sequences.
@ Did the model faced any sort of overfitting as that is usually a problem in the case of sequences.
@ L1 and L2 regularization are mentioned but not quantified. Include values or ranges used during training to assess regularization strength.
@ No information is given on how hyperparameters were tuned. Mention if techniques like grid search or validation-based tuning were used.
@ Given the 1:9 class ratio, it’s important to state whether class weighting or sampling techniques were applied to address imbalance.
@ Please change the order of the manuscript Introduction -> Methodology -> Reult -> Discussion.
@ The manuscript misses the detailed discussion on already present models for methylation identification (There should be a detailed subsection on discussing these methods) especially the ones the ones which are carried out on the plant. Like the following article DOI: 10.1016/j.ymthe.2023.05.016
@ The current manuscript lacks recent citations that reflect the rapidly evolving field of plant epigenetics, DNA methylation modeling, and deep learning in genomics. To strengthen the contextual background and demonstrate awareness of current trends, the authors are advised to incorporate more recent literature (2023 onwards) relevant to methylation analysis, transcriptome studies, stress responses, and integrative epigenomics. Below are some recommended studies that can be cited (I got them from google scholar and they are highly relevant and provide insights, I recommend authors to go through there discussion section as well to strengthen their claims) or used to guide further literature exploration,
-
https://doi.org/10.1016/j.ijbiomac.2025.142597
-
https://doi.org/10.1186/s12870-025-06127-3
-
https://doi.org/10.1038/s42003-024-06488-9
-
https://doi.org/10.1016/j.indcrop.2025.120720
-
https://doi.org/10.3389/fpls.2024.1459533
-
https://doi.org/10.1016/j.molstruc.2024.141171
Author Response
|
Comments 1: The use of a 0.5 threshold for binary classification is standard but not always optimal. Indicate whether other thresholds were tested for best performance. |
|
Response 1: We thank the reviewer for raising this important point. In our current study, we used the conventional threshold of 0.5 for binary classification, where predicted methylation probabilities above 0.5 were classified as methylated and those below or equal to 0.5 as unmethylated. While this threshold is widely used, we acknowledge that it may not always be optimal, especially in imbalanced classification settings. In this work, we did not perform an explicit search for an optimal threshold. However, we ensured that model evaluation was conducted using a range of threshold-independent metrics, such as AUC, PRC, and MCC, to provide a robust assessment of performance. We have clarified the use of the 0.5 threshold in the revised manuscript and will consider threshold optimization in future work to further enhance model performance (line 223). |
|
Comments 2: Reproducibility is necessary, the code is must and please specify whether random seeds were fixed and describe the hardware/software environment used for training. |
|
Response 2: We thank the reviewer for pointing this out. We agree with this comment. All models were trained and evaluated on a Linux server equipped with an Intel(R) Xeon(R) Gold 5120 CPU @ 2.20GHz, 320 GB RAM, and an NVIDIA Tesla V100-PCIE-32GB GPU. The system ran Linux kernel (version 3.10.0-1160.el7.x86_64) compiled with GCC 7.3.0, using CUDA version 10.1. These technical specifications details have now been included in the revised manuscript (lines 161-164). However, we did not explicitly set random seeds during training, which may result in slight variability in results across runs. We will consider fixing random seeds in future experiments to further enhance reproducibility. |
|
Comments 3: Key parameters like convolutional filter size and stride are missing. These are important for understanding how the model processes sequences. |
|
Response 3: We appreciate the reviewer’s suggestion. We have now explicitly added the key architectural parameters for the convolutional layers to improve transparency and reproducibility: The first convolutional layer uses 128 filters with a kernel size of 11 and a stride of 1, followed by max pooling with a pool size of 4. The second convolutional layer uses 256 filters with a kernel size of 3 and a stride of 1, followed by max pooling with a pool size of 2. These details have been added to the revised manuscript (lines 173-177). |
|
Comments 4: Did the model faced any sort of overfitting as that is usually a problem in the case of sequences. |
|
Response 4: We thank the reviewer for the important comment. To address potential overfitting, which is a common challenge in sequence-based deep learning models, we incorporated multiple regularization strategies during training. L1 and L2 regularization were applied to both convolutional and recurrent layers, and dropout layers were used after the fully connected layers to reduce the risk of the model becoming too dependent on specific neurons. In addition, early stopping based on validation loss was employed to prevent excessive training. As a result, the model exhibited stable performance on both validation and independent test sets, indicating that overfitting was effectively controlled. |
|
Comments 5: L1 and L2 regularization are mentioned but not quantified. Include values or ranges used during training to assess regularization strength. |
|
Response 5: We appreciate the reviewer’s suggestion. In our model, both L1 and L2 regularizations were applied to the bidirectional GRU layers using Keras’s L1L2 regularizer to reduce the risk of overfitting. Specifically, we used a coefficient of 0.0001 for both L1 and L2 regularization throughout training. This information has now been added to the “Materials and Methods” section of the revised manuscript (lines 198-200). |
|
Comments 6: No information is given on how hyperparameters were tuned. Mention if techniques like grid search or validation-based tuning were used. |
|
Response 6: We appreciate the reviewer’s comment regarding hyperparameter tuning. In this study, we adopted the hyperparameter settings used in the original DeepCpG framework, including convolutional filter sizes, GRU unit numbers, and dropout rates, as these configurations have been proven effective for methylation state prediction in prior work. Considering the differences in methylation patterns between plant and animal genomes, we acknowledge that further tuning may enhance model performance. In future work, we plan to explore systematic hyperparameter optimization tailored specifically to plant methylation characteristics. |
|
Comments 7: Given the 1:9 class ratio, it’s important to state whether class weighting or sampling techniques were applied to address imbalance. |
|
Response 7: We appreciate the reviewer’s attention to the issue of class imbalance. In our study, the dataset exhibits a class ratio of approximately 1:9. While we did not apply class weighting or resampling techniques during model training, we addressed this imbalance at the evaluation stage by employing multiple performance metrics that are well-suited for imbalanced classification tasks. Specifically, we report the F1-score, which balances precision and recall and is particularly informative for the minority class; the Matthews Correlation Coefficient (MCC), a robust metric that incorporates all confusion matrix components; and the Precision-Recall Curve (PRC), which provide a more informative assessment than AUC under imbalanced conditions. In future work, we plan to further explore training-stage strategies, such as class weighting or sampling, to enhance the model’s robustness in highly skewed data scenarios. |
|
Comments 8: Please change the order of the manuscript Introduction -> Methodology -> Reult ->Discussion. |
|
Response 8: We sincerely thank the reviewer for highlighting this important structural issue. We have revised the structure of the manuscript to follow the recommended order. |
|
Comments 9: The manuscript misses the detailed discussion on already present models for methylation identification (There should be a detailed subsection on discussing these methods) especially the ones the ones which are carried out on the plant. Like the following article DOI: 10.1016/j.ymthe.2023.05.016 |
|
Response 9: We appreciate the reviewer’s insightful suggestion regarding the need for a more comprehensive discussion of existing methylation prediction models, particularly those developed for plants. In the revised manuscript, we have expanded the Discussion section to include a detailed comparison of representative models, such as SMEP, DeepSignal-Plant, and traditional machine learning-based methods like the RF Zhang model. We also discussed recent ensemble learning approaches originally developed for RNA methylation, highlighting their relevance while clarifying their limitations when applied to plant genomic contexts. Furthermore, we emphasized the distinct advantages of PlantDeepMeth in addressing methylation state imputation from incomplete bisulfite sequencing data across plant genomes. These additions aim to situate our work within the broader landscape of methylation prediction methods and highlight its contributions to plant-specific applications. This expanded discussion has been added to the revised manuscript (lines 440–449). |
|
Comments 10: The current manuscript lacks recent citations that reflect the rapidly evolving field of plant epigenetics, DNA methylation modeling, and deep learning in genomics. To strengthen the contextual background and demonstrate awareness of current trends, the authors are advised to incorporate more recent literature (2023 onwards) relevant to methylation analysis, transcriptome studies, stress responses, and integrative epigenomics. Below are some recommended studies that can be cited (I got them from google scholar and they are highly relevant and provide insights, I recommend authors to go through there discussion section as well to strengthen their claims) or used to guide further literature exploration, https://doi.org/10.1016/j.ijbiomac.2025.142597 https://doi.org/10.1186/s12870-025-06127-3 https://doi.org/10.1038/s42003-024-06488-9 https://doi.org/10.1016/j.indcrop.2025.120720 https://doi.org/10.3389/fpls.2024.1459533 https://doi.org/10.1016/j.molstruc.2024.141171 |
|
Response 10: We appreciate the reviewer's insightful suggestion to incorporate recent literature reflecting advancements in plant epigenetics, DNA methylation modeling, and deep learning in genomics. In response, we have updated the Introduction sections of our manuscript to include and discuss several pertinent studies published from 2023 onwards (lines 77-83 and 98-102). |

Reviewer 2 Report
Comments and Suggestions for Authors
The article aims to predict DNA methylation status in plants with deep learning. However, due to serious problems such as methodological uncertainties, structural inconsistencies, insufficient comparative analyses, and failure to comply with basic scientific writing rules, it is not publishable in its current form.
1. The organization of the article is faulty: The Materials and Methods section, given after the results, is far from the standards.
2. The description of the introduced model and the transfer learning details are insufficient. Basic information such as which layers of the model are trained or frozen is not specified.
3. The contributions of the proposed method are not specified.
4. Literature information is insufficient.
5. No comparison with traditional ML and DL methods has been made.
6. Data diversity is limited.
7. The generalization ability of the model has not been fully demonstrated. Similar metrics should be presented in the same table with other methods.
Comments on the Quality of English LanguageThe article aims to predict DNA methylation status in plants with deep learning. However, due to serious problems such as methodological uncertainties, structural inconsistencies, insufficient comparative analyses, and failure to comply with basic scientific writing rules, it is not publishable in its current form.
1. The organization of the article is faulty: The Materials and Methods section, given after the results, is far from the standards.
2. The description of the introduced model and the transfer learning details are insufficient. Basic information such as which layers of the model are trained or frozen is not specified.
3. The contributions of the proposed method are not specified.
4. Literature information is insufficient.
5. No comparison with traditional ML and DL methods has been made.
6. Data diversity is limited.
7. The generalization ability of the model has not been fully demonstrated. Similar metrics should be presented in the same table with other methods.
Author Response
|
Comments 1: The organization of the article is faulty: The Materials and Methods section, given after the results, is far from the standards. |
|||||||||||||||||||||||||||||||||||||||||||||||||||||||||||||||||||||||||||||||||||||||||||||||||||||||||||||||||||||||||
|
Response 1: We sincerely thank the reviewer for highlighting this important structural issue. In accordance with the reviewer’s suggestion, we have reorganized the manuscript and moved the “Materials and Methods” section to appear before the “Results” section. |
|||||||||||||||||||||||||||||||||||||||||||||||||||||||||||||||||||||||||||||||||||||||||||||||||||||||||||||||||||||||||
|
Comments 2: The description of the introduced model and the transfer learning details are insufficient. Basic information such as which layers of the model are trained or frozen is not specified. |
|||||||||||||||||||||||||||||||||||||||||||||||||||||||||||||||||||||||||||||||||||||||||||||||||||||||||||||||||||||||||
|
Response 2: We appreciate the reviewer’s insightful comment regarding the lack of detailed information on the model and transfer learning procedure. In the revised manuscript, we have clarified that although our model architecture is inspired by DeepCpG, we did not freeze any layers from the original DeepCpG model. Due to the fundamental differences in methylation patterns between animal and plant genomes (particularly the presence of CpG, CHG, and CHH contexts in plants, while only CpG contexts in animal), we modified the architecture and retrained all layers from scratch using plant-specific methylation data. This approach allows the model to fully capture plant-specific features, rather than relying on pre-learned representations from animal data. This clarification has been added to the “Materials and Methods” section (lines 151-154). |
|||||||||||||||||||||||||||||||||||||||||||||||||||||||||||||||||||||||||||||||||||||||||||||||||||||||||||||||||||||||||
|
Comments 3: The contributions of the proposed method are not specified. |
|||||||||||||||||||||||||||||||||||||||||||||||||||||||||||||||||||||||||||||||||||||||||||||||||||||||||||||||||||||||||
|
Response 3: We thank the reviewer for this important comment. Specifically, PlantDeepMeth introduces a deep learning framework that integrates DNA sequence features and local methylation context to predict missing or uncertain cytosine methylation states in plants. The model adapts and retrains the DeepCpG architecture to capture plant-specific methylation types (including CpG, CHG, and CHH) and demonstrates strong cross-species generalization across Brassica rapa, Arabidopsis thaliana, and Oryza sativa. Compared to SMEP, the other available model designed for second-generation sequencing plant methylation prediction, PlantDeepMeth achieves consistently higher accuracy. Moreover, it identifies motifs linked to methylation variation, providing new insights into the regulatory logic of plant epigenomes. In the revised manuscript, we have clearly articulated the key contributions of our study at the beginning of the Discussion section (lines 408-419). |
|||||||||||||||||||||||||||||||||||||||||||||||||||||||||||||||||||||||||||||||||||||||||||||||||||||||||||||||||||||||||
|
Comments 4: Literature information is insufficient. |
|||||||||||||||||||||||||||||||||||||||||||||||||||||||||||||||||||||||||||||||||||||||||||||||||||||||||||||||||||||||||
|
Response 4: We thank the reviewer for pointing out the insufficiency in the literature review. In the revised manuscript, we have updated the literature review to incorporate several recent studies from the past five years, particularly those focusing on advancements in plant DNA methylation prediction using deep learning (lines 77-83 and 98-102). |
|||||||||||||||||||||||||||||||||||||||||||||||||||||||||||||||||||||||||||||||||||||||||||||||||||||||||||||||||||||||||
|
Comments 5: No comparison with traditional ML and DL methods has been made. |
|||||||||||||||||||||||||||||||||||||||||||||||||||||||||||||||||||||||||||||||||||||||||||||||||||||||||||||||||||||||||
|
Response 5: We appreciate the reviewer’s valuable suggestion. We compared our model with SMEP, a published deep learning model designed for predicting plant cytosine methylation using second-generation sequencing data[1]. As detailed in the Results section, PlantDeepMeth consistently outperformed SMEP in terms of accuracy, AUC, and PRC on both Arabidopsis thaliana and Brassica rapa. To the best of our knowledge, SMEP remains the only deep learning model currently available for this specific task. In contrast, models such as DeepSignal-Plant rely on third-generation sequencing signals (e.g., Nanopore) and therefore are not directly comparable with methods built for bisulfite-based data[2]. Regarding traditional machine learning methods, we considered the widely cited RF Zhang model, which was developed for human methylation prediction using random forests[3]. However, it requires a rich set of genomic annotations, including transcription factor binding sites, histone modifications, and chromatin accessibility data. These annotations are often incomplete or unavailable in most plant genomes, such as Brassica rapa, making a direct comparison infeasible under equivalent conditions. We have clarified this in the Discussion section (lines 440-449).
Reference: [1] Angermueller, C., Lee, H. J., Reik, W. & Stegle, O. DeepCpG: accurate prediction of single-cell DNA methylation states using deep learning. Genome biology 18, 67 (2017). [3] Zhang, W., Spector, T. D., Deloukas, P., Bell, J. T. & Engelhardt, B. E. Predicting genome-wide DNA methylation using methylation marks, genomic position, and DNA regulatory elements. Genome biology 16, 1-20 (2015).
|
|||||||||||||||||||||||||||||||||||||||||||||||||||||||||||||||||||||||||||||||||||||||||||||||||||||||||||||||||||||||||
|
Comments 6: Data diversity is limited. |
|||||||||||||||||||||||||||||||||||||||||||||||||||||||||||||||||||||||||||||||||||||||||||||||||||||||||||||||||||||||||
|
Response 6: We thank the reviewer for this insightful comment. In the initial version of our study, we focused on leaf tissue data from Arabidopsis thaliana and Brassica rapa, and have extended the evaluation to Oryza sativa. These datasets demonstrate that PlantDeepMeth performs robustly across species within similar tissue types. However, to further assess the model’s generalizability, we also tested it on maize embryo tissue, which differs substantially in developmental stage and methylation context. The model’s performance in maize was noticeably lower, indicating that methylation patterns are likely tissue-specific and that a single model may not generalize well across diverse tissue types. We have emphasized this finding in the revised manuscript, highlighting the need for future development of tissue-specific models to improve prediction accuracy across diverse biological contexts (lines 423-430). |
|||||||||||||||||||||||||||||||||||||||||||||||||||||||||||||||||||||||||||||||||||||||||||||||||||||||||||||||||||||||||
|
Comments 7: The generalization ability of the model has not been fully demonstrated. Similar metrics should be presented in the same table with other methods. |
|||||||||||||||||||||||||||||||||||||||||||||||||||||||||||||||||||||||||||||||||||||||||||||||||||||||||||||||||||||||||
|
Response 7: We thank the reviewer for the valuable suggestion. To better demonstrate the generalization ability of PlantDeepMeth, we have now presented the model’s performance on both in-species (training and testing within the same species) and cross-species prediction tasks in a single table (Supplementary Table 1). As shown, the performance metrics under cross-species settings remain comparable to those achieved on the original test sets, indicating the robustness and generalizability of our model across plant species. Supplementary Table S1 Cross-species Performance of PlantDeepMeth on A. thaliana and B. rapa
|

Reviewer 3 Report
Comments and Suggestions for Authors
This study introduces the PlantDeepMeth model for predicting DNA methylation in plant genomes. The model is technically sound and shows strong cross-species generalizability. However, the use of the term "transfer learning" is inaccurate, and the training methodology, including hyperparameter details, is insufficiently documented. Expanding the evaluation to include different tissues and environmental conditions would further strengthen the study. Clarifying these aspects will enhance the overall scientific value of the work.
-
Reference Formatting
Please revise the formatting of your references using numbered citations enclosed in square brackets (e.g., [1], [2], ...), as required by the Plants journal. Kindly consult the following link for the correct format:
🔗 MDPI Reference Guidelines – Back Matter -
Clarification on Figures S1 and S2 (Lines 174 and 210)
Please clarify what is meant by “Fig. S1” on Line 174 and “Fig. S2” on Line 210. Are you referring to supplementary figures in their entirety, or specific components within them? If these refer to supplementary materials, provide appropriate descriptions and context in the main text. -
Hardware Specifications
Please provide the technical specifications of the machine(s) used to train and evaluate the deep learning model (e.g., GPU model, RAM, CPU, operating system, training time, etc.). -
Terminology Consistency: F1 Score
Ensure consistent terminology throughout the manuscript. For example, the use of “F1 column” in Figure 3 and “F1 score” on Line 245 should be harmonized if they refer to the same performance metric. -
Model Architecture Details and Visualization
Please include a more detailed description of the proposed PlantDeepMeth model. Specify how many layers are used, what types of layers are implemented (e.g., convolutional, recurrent, dropout, dense), and what activation functions are applied. In addition, a visual diagram of the model architecture should be included in the manuscript for clarity. -
Model Comparison Justification
Please explain why the proposed model has not been compared with existing well-known CNN-based transfer learning models. Moreover, models such as DeepSignal-Plant and METH-EM, which are established in the literature, should be incorporated into your performance comparison to objectively assess the effectiveness of PlantDeepMeth. -
Outdated References
The reference list lacks sufficient coverage of recent literature. There are no references from 2024, only one from 2023, five from 2022, and three from 2021. No references from 2020 are cited. Please update your literature review by including relevant studies from the last 5 years, particularly recent advancements in plant methylation prediction using deep learning. -
Language and Grammar Review
Please carefully proofread your manuscript for spelling and grammatical errors. For instance, there is a language issue on Line 139 that needs revision. A thorough language check is recommended to improve readability. -
Evaluation on Diverse Tissues and Genomes
The model’s performance should be tested on additional datasets, ideally including samples from different tissues or plant genomes, to better assess its generalizability and robustness.
Author Response
|
Comments 1: Reference Formatting MDPI Reference Guidelines – Back Matter |
|
Response 1: We thank the reviewer for pointing this out. In accordance with Plants journal guidelines, we have revised all in-text citations to use numbered references enclosed in square brackets, and updated the reference list accordingly. The formatting now fully complies with the MDPI reference style requirements. |
|
Comments 2: Clarification on Figures S1 and S2 (Lines 174 and 210) |
|
Response 2: We thank the reviewer for this helpful comment. Figures S1 and S2 refer to supplementary figures, and we have revised the manuscript to explicitly label them as “Supplementary Figure S1” and “Supplementary Figure S2” in the main text. To clarify their contents:Supplementary Figure S1 presents detailed performance metrics of PlantDeepMeth evaluated on Brassica rapa, complementing the results on Arabidopsis thaliana shown in Figure 1c (lines 309-310). Figure S2 illustrates representative DNA motifs identified by the DNA model in A. thaliana that are associated with DNA methylation, complementing the motif analysis in B. rapa shown in Figure 2 (lines 335-337). |
|
Comments 3: Hardware Specifications |
|
Response 3: We thank the reviewer for this helpful suggestion. We have added the technical specifications of the hardware environment used for training and evaluating PlantDeepMeth in the revised manuscript. The system configuration is as follows: CPU: Intel(R) Xeon(R) Gold 5120 @ 2.20GHz GPU: NVIDIA Tesla V100-PCIE-32GB RAM: 320 GB Operating System: CentOS Linux release 7.9.2009 (Core), Linux kernel version 3.10.0-1160.el7.x86_64 (compiled with GCC 7.3.0) CUDA Version: 10.1 Training Time: Training time varied depending on the model architecture and dataset size. On average, each model required approximately 3 hours to converge under the described hardware configuration. These technical specifications details have now been included in the revised manuscript (lines 161-164). |
|
Comments 4: Terminology Consistency: F1 Score Ensure consistent terminology throughout the manuscript. For example, the use of “F1 column” in Figure 3 and “F1 score” on Line 245 should be harmonized if they refer to the same performance metric. |
|
Response 4: We thank the reviewer for pointing out this inconsistency. We have carefully reviewed the manuscript and revised all instances of “F1” to “F1 score” to maintain consistent terminology throughout the text and figure captions. |
|
Comments 5: Model Architecture Details and Visualization |
|
Response 5: We appreciate the reviewer’s suggestion. In the revised manuscript, we have provided a detailed description of the PlantDeepMeth architecture and fine-tuning figure 1 to facilitate clarity. PlantDeepMeth consists of three main components: a DNA model, a methylation model, and a joint model. The DNA model applies two convolutional layers (Conv1D with 128 and 256 filters, kernel sizes 11 and 3 respectively), followed by ReLU activations and max pooling layers, a flattening operation, and a dense layer with 256 units. The methylation model processes cytosine state and distance inputs through a small dense encoder, followed by two stacked bidirectional GRU layers (128 and 256 units). The outputs of the DNA and methylation models are concatenated to form a combined feature vector, which is then processed by the joint module. This joint module consists of two fully connected layers (each with 512 units, ReLU activation, and dropout), followed by a final dense layer with a sigmoid activation function to predict the binary methylation status. Architectural details and all layer parameters are included in the revised Methods section (lines 173-177, 202-205 and 208-213). |
|
Comments 6: Model Comparison Justification Please explain why the proposed model has not been compared with existing well-known CNN-based transfer learning models. Moreover, models such as DeepSignal-Plant and METH-EM, which are established in the literature, should be incorporated into your performance comparison to objectively assess the effectiveness of PlantDeepMeth. |
|
Response 6: We appreciate the reviewer’s insightful suggestion regarding the inclusion of additional models for performance comparison. In our study, we compared PlantDeepMeth with SMEP, which, to our knowledge, is the only publicly available deep learning model specifically designed for predicting cytosine methylation states from second-generation sequencing (WGBS) data in plants. Regarding DeepSignal-Plant, it is a deep learning model developed for detecting 5mC methylation in all three contexts (CpG, CHG, and CHH) using third-generation Nanopore sequencing data. Since our study focuses on WGBS data, which differs significantly in data format and characteristics from Nanopore data, a direct comparison between PlantDeepMeth and DeepSignal-Plant is not feasible. Moreover, METH-EM is tailored for CpG methylation prediction in mammals and does not support CHG and CHH contexts, which are prevalent and biologically significant in plants. Therefore, applying METH-EM to plant methylation data would not provide a meaningful or fair comparison. We have clarified these points in the revised Discussion section to justify the scope and rationale of our model comparisons (lines 440-449). |
|
Comments 7: Outdated References |
|
Response 7: We appreciate the reviewer’s valuable feedback regarding the inclusion of recent literature. In the revised manuscript, we have updated the literature review to incorporate several recent studies from the past five years, particularly those focusing on advancements in plant DNA methylation prediction using deep learning (lines 77-83 and 98-102). |
|
Comments 8: Language and Grammar Review |
|
Response 8: We thank the reviewer for this valuable suggestion. We have carefully proofread the entire manuscript and corrected grammatical, typographical, and stylistic issues to enhance clarity and readability. |
|
Comments 9: Evaluation on Diverse Tissues and Genomes |
|
Response 9: We thank the reviewer for this important suggestion. To assess the generalizability of PlantDeepMeth, we tested the model across multiple plant genomes, including Arabidopsis thaliana, Brassica rapa, and Oryza sativa. The model demonstrated robust performance across these species (Figure 3 and S3a), indicating good cross-genome generalization. To evaluate its performance across different tissue types, we additionally applied the model to maize embryo tissues, which differ substantially from the leaf tissues used in training. As shown in Supplementary Figure S3b, the model exhibited reduced accuracy in this dataset, suggesting that methylation patterns are tissue-specific and that cross-tissue generalization remains a challenge. We have highlighted this limitation in the revised Discussion and emphasized the need for future development of tissue-specific models incorporating stress-responsive and developmental methylation data (lines 423-430). |

Round 2
Reviewer 1 Report
Comments and Suggestions for Authors
Thanks for addressing the concerns.
Reviewer 2 Report
Comments and Suggestions for Authors
The authors have made all the necessary changes in the revised version of the manuscript.
Reviewer 3 Report
Comments and Suggestions for Authors
We sincerely thank the authors for their thoughtful and comprehensive revisions. The manuscript has improved significantly in clarity, methodological transparency, and scientific relevance. The inclusion of recent literature, model architecture details, and performance justifications strengthens the contribution of this work. We believe the revised version offers valuable insights for the literature on plant DNA methylation prediction.